# A Method for the Automatic Extraction of Support Devices in an Overhead Catenary System Based on MLS Point Clouds

Shengyuan Zhang [1], Qingxiang Meng [1,*], Yulong Hu [2], Zhongliang Fu [1] and Lijin Chen [3]

1   School of Remote Sensing and Information Engineering, Wuhan University, Wuhan 430079, China
2   China Transport Telecommunications & Information Center, Beijing 100011, China
3   Center for Health Statistics and Information, National Health Commission of the People's Republic of China, Beijing 100044, China
*   Correspondence: mqx@whu.edu.cn

**Abstract:** A mobile laser scanning (MLS) system can acquire railway scene information quickly and provide a data foundation for regular railway inspections. The location of the catenary support device in an electrified railway system has a direct impact on the regular operation of the power supply system. However, multi-type support device data accounts for a tiny proportion of the whole railway scene, resulting in its poor characteristic expression in the scene. Therefore, using traditional point cloud filtering or point cloud segmentation methods alone makes it difficult to achieve an effective segmentation and extraction of the support device. As a result, this paper proposes an automatic extraction algorithm for complex railway support devices based on MLS point clouds. First, the algorithm obtains hierarchies of the pillar point clouds and the support device point clouds in the railway scene through high stratification and then realizes the noise that was point-cloud-filtered in the scene. Then, the center point of the pillar device is retrieved from the pillar corridor by a neighborhood search, and then the locating and initial extracting of the support device are realized based on the relatively stable spatial topological relationship between the pillar and the support device. Finally, a post-processing optimization method integrating the pillar filter and the voxelized projection filter is designed to achieve the accurate and efficient extraction of the support device based on the feature differences between the support device and other devices in the initial extraction results. Furthermore, in the experimental part, we evaluate the treatment effect of the algorithm in six types of support devices, three types of support device distribution scenes, and two types of railway units. The experimental results show that the average extraction IoU of the multi-type support device, support device distribution scenes, and railway unit were 97.20%, 94.29%, and 96.11%, respectively. In general, the proposed algorithm can achieve the accurate and efficient extraction of various support devices in different scenes, and the influence of the algorithm parameters on the extraction accuracy and efficiency is elaborated in the discussion section.

**Keywords:** overhead catenary system support device; mobile laser scanning; hierarchical chunking; target extraction; integrated filter

## 1. Introduction

The efficient automatic extraction of railway device data is critical to developing railway digital twins [1,2]. Currently, this is accomplished by constructing a railway digital twin [3] model based on mobile measuring equipment, such as mobile laser scanning (MLS) [4], and then realizing the regular inspection of the railway system by measuring the geometric parameters of railway device models [5,6]. A support device is widely employed in railway electrification projects as a bearing device of the railway power supply system [7,8]. However, support devices in various railway scenes significantly differ in various types of systems while accounting for only a small proportion of the whole scene (Figure 1a). However, the conductive height and pull-value of key railway parameters can

be calculated based on the relevant data of the support devices (Figure 1b). As a result, the fully automatic, efficient, and accurate extraction of support devices in railway scenes is of great value and significance.

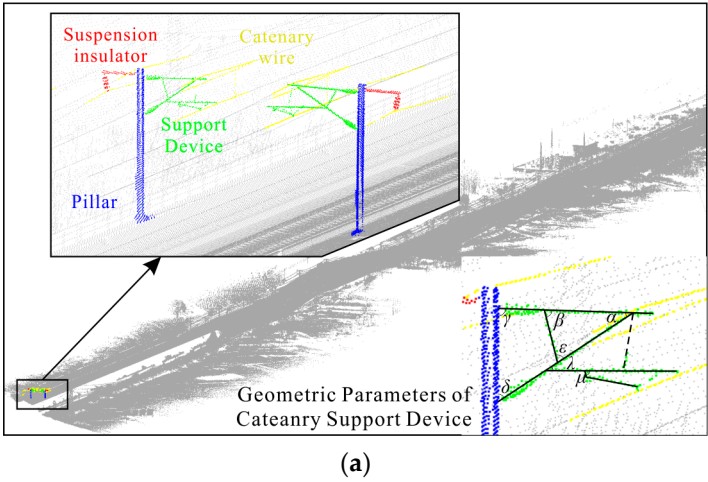 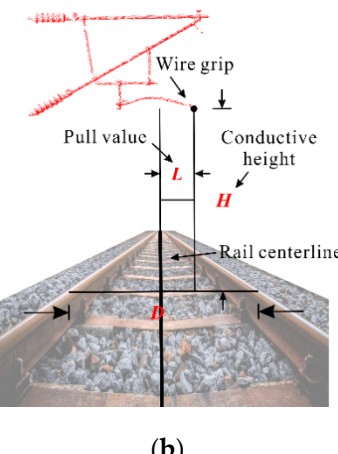

(**a**) (**b**)

**Figure 1.** Support device in a complex railway scene and its geometric parameter measurement. (**a**) A support in a complex railway scene. (**b**) The measurement method for some geometric parameters of a railway system.

Extracting a support device in a railway scene is part of object extraction research with the point cloud. As a result, this paper seeks solutions to support device extraction by discussing and analyzing relevant research on specific object extraction methods in point cloud scenes [9,10]. Some studies have acquired railway scene support device information using photogrammetry [7]. However, the maintenance of railway line-related devices is often carried out at night, which makes illumination inevitably affect the collected data. MLS technology can efficiently obtain massive point cloud scene data and is not affected by lighting conditions. Currently, the solutions for object extraction in point cloud scenes can be divided into point cloud filtering [11] and point cloud segmentation [12,13].

Filtering non-target points while retaining target points is what point cloud filtering is all about [14]. It primarily includes the statistical-based [15,16], neighborhood-based [17], projection-based [18], signal-processing [19], and hybrid point-cloud-filtering [20] methods. These methods filter noise points by distinguishing feature differences between target and non-target objects and are appropriate for targets with apparent features in simple scenes [21]. However, due to the interference of other railway devices in the scene, the point cloud filtering method has difficultly in directly capturing the features of the support device in the railway scene. At the same time, the differences in feature expression of different types of support devices also increases the processing difficulty of the point cloud filtering method. It is not easy to effectively extract support devices in complex railway scenes through only a point cloud filtering algorithm.

The point cloud segmentation method achieves target object segmentation and extraction by dividing the point cloud scene into several mutually exclusive subsets. It includes the edge-based [22,23], region-growing [24], model-fitting [25], and clustering-based [26] methods. The edge-based segmentation method obtains edge information by judging the changes in point vectors and then realizing the target object extraction in the scene. The algorithm is simple in structure and has a good segmentation effect on the target with the apparent edge feature. The clustering-based method assigns points with similar feature distributions to corresponding categories, and then the specific target of the point cloud can be clustered [27,28]. However, these two segmentation methods are unsupervised clustering methods, and it is challenging to segment the support device in the railway scene selectively. The model fitting method achieves target segmentation and extraction by matching and recognizing the point cloud data and the target geometry model [29].

Although this method has a good segmentation effect on objects with regular geometry, it takes a lot of time to fit the multi-type support device models. The region-growing segmentation method combines the seed points [30] with the same feature points in the neighborhood to achieve point cloud segmentation for various objects. Although this method can obtain good edge information and target segmentation results, its clustering effect is closely related to the selection of seed points [31]. In a nutshell, the uncertainty of the distribution of the support device in the scene [32,33], as well as the weak feature expression of the small data scale in the scene [34], make it difficult for existing point cloud segmentation methods to be directly applied to the extraction process of a railway scene support device.

Based on the above questions, an automatic extraction method of the support device based on the MLS point cloud is proposed and is based on the characteristics of the support device. To begin, to effectively solve the influence caused by the shadowing effect of the railway background point cloud, this study filters out the noise points by hierarchical chunking and realizes the division of the multi-batch point cloud units. Then, to further amplify the expressive features of the support device, the support device is located and extracted based on the relatively stable spatial relationship between the support device and the pillar and the pillar center's information. Finally, the fine extraction work of the support device is completed by optimizing the post-processing of the initial extraction support device by integrating the pillar filter and the voxelized projection filter. To summarize, the main contributions of this paper are as follows:

(1)    A new method is proposed for locating support devices based on relatively stable spatial relationships between railway devices. Because each support device has a pillar center point, combining the two retrievals can reduce the occurrence of missing support devices and repeated extraction.

(2)    To achieve the high-precision extraction of the support device, other railway devices in the initial extraction results are filtered out by integrating two filters, the pillar and the voxel projection, which significantly improves the extraction accuracy of the support device. Among them, the voxel scale of the voxel projection filter is re-analyzed and designed based on the characteristics of the contact wires in the scene.

(3)    To assess the extraction effect and robustness of the proposed algorithm, six types of support devices and three types of support device distribution scenes are tested. Furthermore, two groups of railway unit scenes are tested to detect the performance of the algorithm in the actual application process.

The manuscript is organized as follows: The proposed approach is explained in Section 2. Section 3 analyzes and tests the relevant parameters and algorithm performance and demonstrates the effectiveness and robustness of the algorithm. In addition, the ablation experiment also discusses each integrated filter component's characteristics. The last section provides a synthesis of conclusions and our main contributions.

## 2. Method

Using the MLS point cloud, the proposed method can extract the support devices automatically and accurately using a gradual processing scheme, including the scene hierarchical chunking, positioning and initial extraction of the support device, and result optimization process. All these steps make a difference in the proposed method's high extraction accuracy. The algorithm flow chart is shown in Figure 2. Firstly, the original railway scene is divided into hierarchical chunking based on key trajectory points. Affine transformation is performed on the divided region blocks to facilitate batch processing of the railway scenes. Then, the support device is located through the appropriate pillar center point of the neighborhood search and the relatively stable spatial relationship between the pillar and the support device, and then the initial extraction of the support device is realized. Finally, the pillar and voxelized projection filters are integrated to filter out the pillars, suspended insulators, and contact wires in the initial extraction results, and

the high-precision automatic extraction of support devices in complex railway scenes is gradually realized.

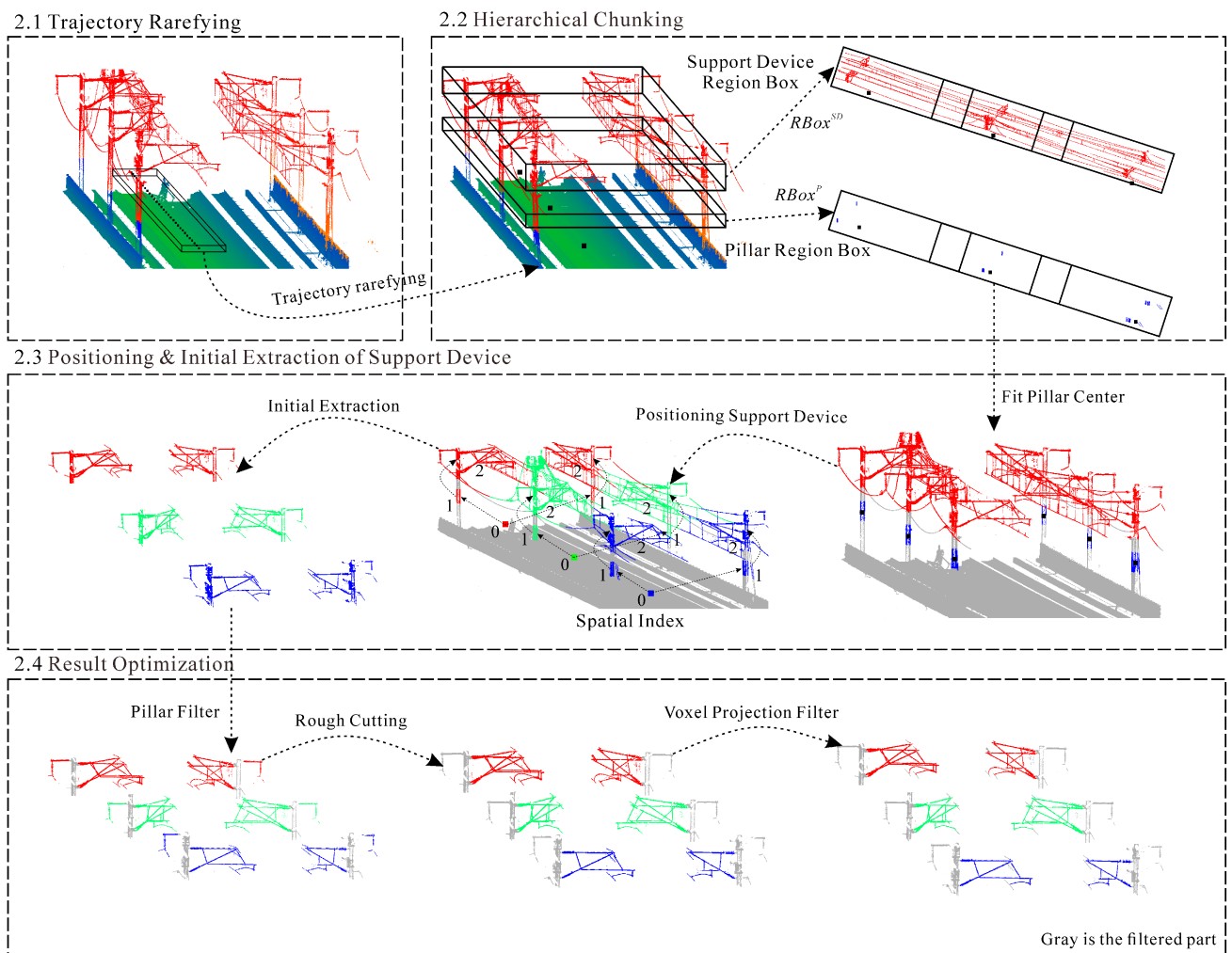

**Figure 2.** The overall flowchart of the algorithm.

### 2.1. Trajectory Rarefying

It is difficult to directly process railway scenes because of their large data scales, long spans, and numerous curve changes. Therefore, we have divided them into a series of railway scene units with the same span and with partial overlap based on the trajectory points $P^L$ to enable batch processing of the original railway scene $P^C$, defined as: $P^L = \left\{ p_j^L \middle| j = 1, 2, \dots, n^{P^L} \right\}$, $P^C = \left\{ p_i^C \middle| i = 1, 2, \dots, n^{P^C} \right\}$, where $n^{P^L}$ and $n^{P^c}$ are the total amounts of $P^L$ and $P^C$, respectively, and $p_i^C$ and $p_j^L$ represent points in $P^C$ and $P^L$, respectively.

Although the close trajectory spacing in the original trajectory data makes it easier to process the railway scene units with small spans, it also results in more support devices in the scene being divided into multiple railway scene units. Assuming the trajectory thinning threshold is set to $\psi$, the trajectory point can be divided into $\lfloor N/\psi \rfloor + 1$ ($\lfloor x \rfloor$ means $x$ rounded down) unit point sets, and we can define the total point set as $P^T = \left\{ p_k^T | k = 1, 2, \dots, n^{P^T} \right\}$, where $n^{P^T}$ is the total amount of $P^T$. Each center point of the unit is designated as a key trajectory point. The last $P^T$ contains $N\%\psi$ points and the relationship between $\psi/2$ and $N\%\psi$ determines whether the unit set has a data value for

batch-processing railway scenes. Figure 3 depicts a schematic diagram of trace drainage, where $\boldsymbol{P}^{KTCP}$ is calculated as a set of $p^{KTCP}$ using Formula (1):

$$
\boldsymbol{P}^{KTCP} = \begin{cases} \frac{1}{\psi}\sum\limits_{k=1}^{\psi} p_k^T & , n^{\boldsymbol{P}^T} = \psi \\ \frac{1}{N\%\psi}\sum\limits_{k=1}^{N-N\%\psi} p_k^T & , \psi/2 \le n^{\boldsymbol{P}^T} < \psi \\ \text{Discarded} & , n^{\boldsymbol{P}^T} < \psi/2 \end{cases}
\tag{1}
$$

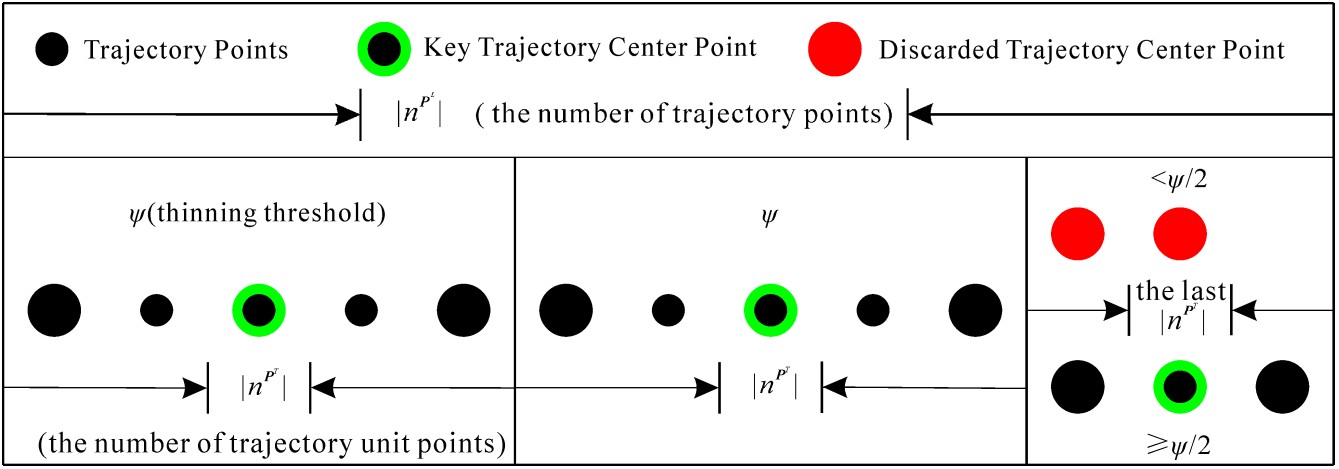

**Figure 3.** Schematic diagram of trajectory extraction.

### 2.2. Hierarchical Chunking

There are many noise point clouds in a railway scene whicsignificantly impact the extraction accuracy and efficiency of support devices. As a result, to achieve the hierarchical processing of point cloud scenes, this paper constructs corresponding data corridors of pillars and support devices based on the significant differences in the spatial locat h ions of the pillars and support devices. Furthermore, to realize the batch processing of the scene point cloud, the dataset of key trajectory points is used as the data reference and performs the chunking process on the data corridor of the pillar and support device. In particular, to amplify the feature expression of the support device during subsequent processing, we first defined and built a pillar region block ($RBox^P = \{l, w, h_1\}$) and support device region block ($RBox^{SD} = \{l, w, h_2\}$) around each key trajectory point. Furthermore, to avoid additional noise points caused by the inconsistency between the region block's attitude and the orbit's direction, we constructed the rotation matrices of $\mathbf{RMat}^\beta$ and $\mathbf{RMat}^\alpha$ according to the offset angle $\beta$ and $\alpha$ between the direction of the orbit and the X-axis and Z-axis, respectively. Then, the attitude of $RBox^P$ and $RBox^{SD}$ were adjusted by Formulas (3) and (2), respectively, to reduce the adverse effects of the terrain relief and curved orbit on the extraction process.

$$
\alpha = \arccos\left(\frac{x_{p^{KTCP}} - x_{\boldsymbol{P}_i^L}}{\left|\overrightarrow{p^{KTCP}\boldsymbol{P}_i^L}\right|}\right)
\tag{2}
$$

$$
\beta = \arccos\left(\frac{z_{p^{KTCP}} - z_{\boldsymbol{P}_i^L}}{\left|\overrightarrow{p^{KTCP}\boldsymbol{P}_i^L}\right|}\right)
\tag{3}
$$

$$RBox'(x',y',z') = RBox(x,y,z) \cdot \mathbf{RMat}^\alpha \cdot \mathbf{RMat}^\beta \tag{4}$$

Among them: $\mathbf{RMat}^\alpha = \begin{bmatrix} \cos\alpha & -\sin\alpha & 0 \\ \sin\alpha & \cos\alpha & 0 \\ 0 & 0 & 1 \end{bmatrix}$. $\mathbf{RMat}^\beta = \begin{bmatrix} \cos\beta & 0 & -\sin\beta \\ 0 & 1 & 0 \\ \sin\beta & 0 & \cos\beta \end{bmatrix}$.

### 2.3. Positioning and Initial Extraction of the Support Device

To further amplify the feature expression of the support device in the processing point clouds, the pillar device and the support device were bonded according to the relatively stable spatial topological relationship between them, and then we realized the positioning and initial extraction of the support device. Figure 4 shows the initial extraction process of the support device. The critical idea was to complete the support device's initial extraction by constructing the support device's initial extraction region block using the support object's spatial location as a reference. Therefore, it was necessary to quickly and accurately extract pillar points in the pillar corridors. An algorithm was designed for the pillar center point based on a neighborhood search to extract the pillar point cloud in the pillar corridor effectively. Specifically, due to the significant distances between the pillars, we chose a random point in the pillar data corridor as a seed point to complete the construction of the neighborhood region blocks ($RBox^N$). Then, by analyzing the number of pillar points in $RBox^N$, a series of initial pillar center points ($p^{IPC}$) was obtained. Furthermore, the pillar detection region block ($RBox^{PI}$) was built on $p^{IPC}$ because the compensation device on the pillar had similar characteristics (in terms of the number of point clouds) to the pillar in the pillar corridors. The pseudo pillar center points could then be filtered out by analyzing the characteristics of the point clouds in $RBox^{PI}$. The detailed procedure is shown in Algorithm 1.

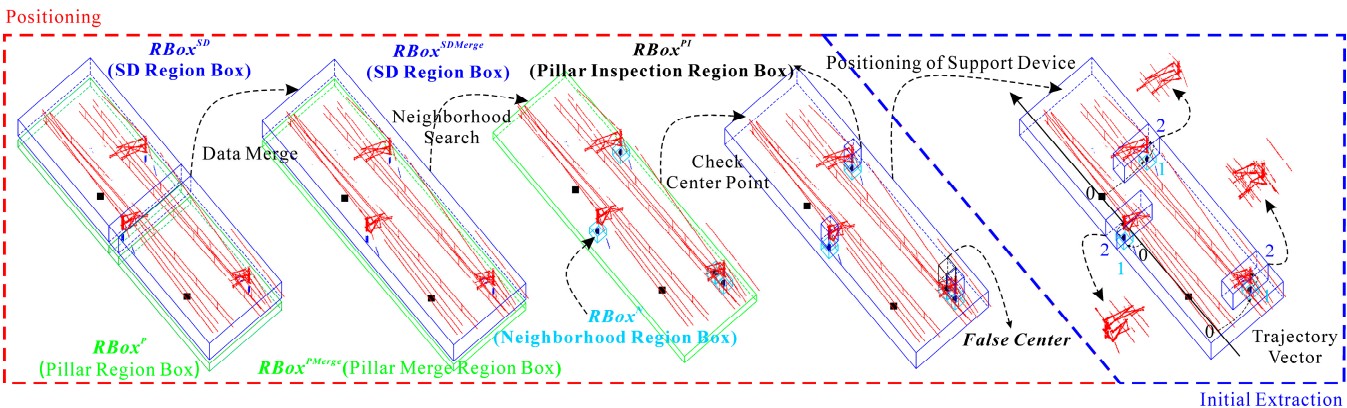

**Figure 4.** Process flow diagram for the initial extraction of the OCS-S.

### 2.4. Result Optimization

To achieve the accurate extraction of the support device, the pillars, suspension insulators, and contact wires were gradually removed by integrating the pillar and voxel projection filter. Figure 5 shows the stepwise optimization process. The pillar exhibits a vertical distribution along a single direction in the railway scene. Therefore, the center point of the neighborhood search was utilized as the data reference and the vertical distribution characteristics of the pillar were used to build the pillar filter, which was applied to filter the pillar device point cloud. The pillar filter was a region block with a length, width, and height of 0.8 m, 2 m, and 4 m, respectively (Figure 5a). Furthermore, the suspension insulator and the support device with contact wire in the initial extraction region block were located in two separate spaces because of the filtering of the pillar. Therefore, the suspension insulator filtering in the initial extraction results would be realized by using the differences in point density in the two regions. Specifically, the scale of the point cloud

of the suspension insulator was far lower than that of the support device with contact wire, and the rough extraction of the support device could be achieved by comparing the selected region block with a large point density (Figure 5b). The difficulty of filtering the contact wire was proportional to the linear distance from the support device. The rough extracted point cloud voxel was first projected to the grid along the Z-axis to filter the contact wire point clouds. Since most of the support device point clouds were located in the same XOZ plane (Figure 5c), the difference in point density between the support device and the contact wire in the projection results would be further widened. Then, the original area was divided into grids whose length and width were $w_0$ and $d$, respectively. The contact wires were filtered in the rough extraction results by judging the point cloud density in the grid to achieve the accurate extraction of the support devices.

---

**Algorithm 1.** Algorithm for extracting pillar center points

| | |
|---|---|
| **Input:** | pillar region box: $\{ RBox^P \mid x \in (x_1, x_2) \cap y \in (y_1, y_2) \cap z \in (z_1, z_2 + h_1) \}$; <br> OCS-SD region box: $\{ RBox^{SD} \mid x \in (x_1, x_2) \cap y \in (y_1, y_2) \cap z \in (z_2, z_2 + h_2) \}$; <br> neighborhood region box: <br> $\{ RBox^N \mid x \in (x_3, x_4) \cap y \in (y_3, y_4) \cap z \in (z_1, z_1 + h_1) \}$; <br> pillar inspection region box: <br> $\{ RBox^{PI} \mid x \in (x_5, x_6) \cap y \in (y_5, y_6) \cap z \in (z_2, z_2 + h_2) \}$; <br> key trajectory center points set: $\boldsymbol{P}^{KTCP} = \left\{ p_i^{KTCP} \mid i = 1, 2, \dots, n^{\boldsymbol{P}^{KTCP}} \right\}$, where $n^{\boldsymbol{P}^{KTCP}}$ is the total amount of $\boldsymbol{P}^{KTCP}$. |
| **Output:** | set of pillar center points: $\boldsymbol{P}^{RPC}$. |

1:  **for** $i = 1$ *to* $n^{\boldsymbol{P}^{KTPC}}$ **do**
2:      initialize $RBox^{PMerge} = RBox_i^P \cup RBox_{i-1}^P \cup RBox_{i+1}^P$;
3:      initialize $RBox^{SDMerge} = RBox_i^{SD} \cup RBox_{i-1}^{SD} \cup RBox_{i+1}^{SD}$;
4:      **for** $j = 0$ *to* $n^{\boldsymbol{P}^C}$ **do**
5:          **if** $(p_j^C \in RBox^N)$ **then**
6:                                                                 initialize $k = 0$;
7:                                                                 $k$++;
8:                                                                 $\boldsymbol{P}^{RBox^N}.\text{AddPoint}\left(p_i^C\right) \leftarrow$ Add to the collection;
9:          **end if**
10:     **end for**
11:     **if** $(k > \delta)$ **then** $\leftarrow \delta$ is the pillar center point initial extraction threshold in $RBox^N$
12:         $p_i^{IPC} = \text{avg}\left( \boldsymbol{P}^{RBox^N} \right) \leftarrow$ Obtain the initial pillar center points
13:     **end if**
14: **end for**
15: **for** $i = 0$ *to* $n^{\boldsymbol{P}^{IPC}}$ **do**
16:     **for** $j = 0$ *to* $n^{\boldsymbol{P}^C}$ **do**
17:         initialize $k = 0$;
18:         **if** $(p_j^C \in RBox^{PI})$
19:                 $k$++;
20:         **end if**
21:         **if** $(k > \lambda) \leftarrow \lambda$ is the pillar center point check threshold in $RBox^{CPt}$
22:             $p_i^{RPC} = p_i^{IPC}$
23:         **end if**
24:     **end for**
25: **end for**
26: **return** $\boldsymbol{P}^{RPC}$;

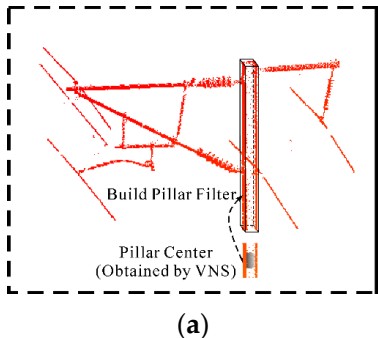 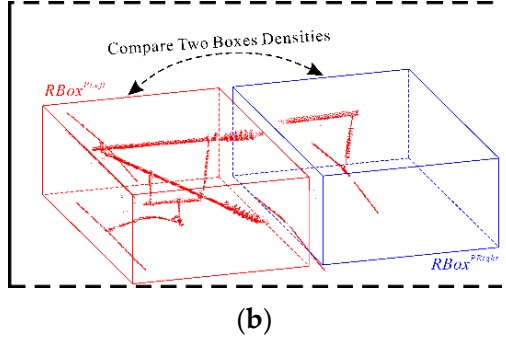 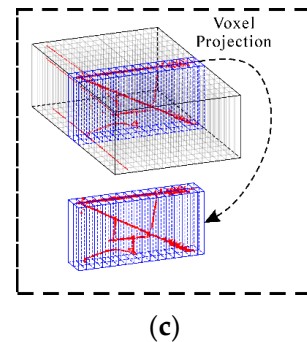

(**a**)          (**b**)          (**c**)

**Figure 5.** Flowchart of the result optimization. (**a**) Pillar filter. (**b**) Rough crop. (**c**) Voxel projection filter.

## 3. Experiments

### 3.1. Study Area and Dataset

The 3D LiDAR dataset collected by the light movement scanning measurement system along the Yancheng–Nantong railway is shown in Figure 6. The system comprises an on-orbit surveying and scanning vehicle and a high-precision laser scanning device, Z + F Profile9012. It takes 2 km as a measurement period to measure the data of two groups of railway units from Yancheng–Nantong, with an average number of points of approximately 200 million. The trajectory data is collected while the MLS system scans the railway scene, which is a critical spatial reference to support device positioning. Among them, the GNSS receiver is integrated with the Z + F Profile 9012 device, and the trajectory positioning accuracy is further improved through differential GPS technology.

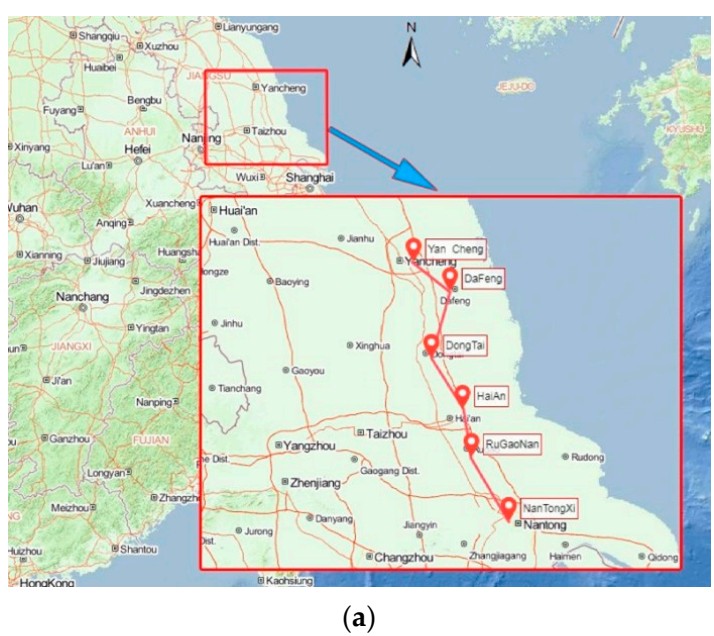

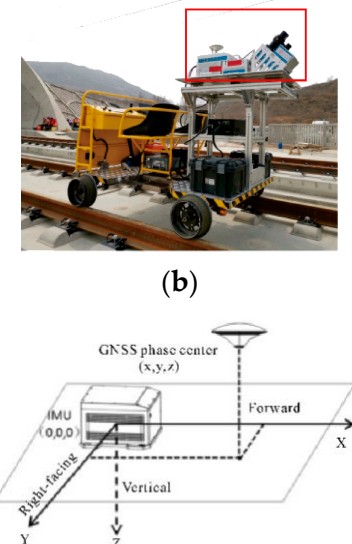

(**a**)          (**b**)

(**c**)

**Figure 6.** Experimental datasets. (**a**) Railway dataset from Yancheng to Nantong. (**b**) Z + F Profile 9012. (**c**) GNSS track acquisition device.

Figure 7 depicts the numerous types of support devices in a railway scene. Distinct types of support devices have different accessories and expression characteristics. Because of this, it was critical to validate the algorithm's robustness through an extensive assessment of the support device's extracts results with various types of devices. Among them, the structures of a single support device (SSD) and a double support device (DSD) are relatively straightforward. The main difference between them is the number of support devices on

the pillar and the pillar type. The single ratchet support device (SRSD) and double ratchet support device (DRSD) build on the original support device with a compensating device to automatically adjust the tension of the contact wire and the bearing cable. As in a DSD structure, steel frame support devices (SFSD) are commonly encountered at railway platforms. The loop support device (LSD) is structurally similar to the SSD, but the critical distinction is that the suspension insulator is in the LSD. Among them, the supply voltage suspension insulator of the LSD is 2 × 25 kV.

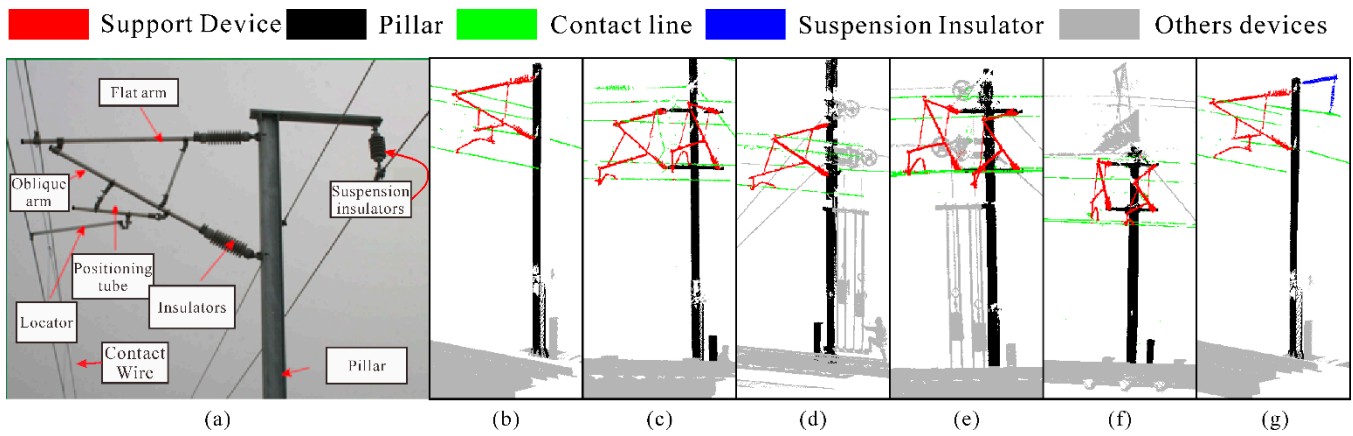

**Figure 7.** Display diagram of the six types of support devices: (**a**) the original image of the support device, (**b**) the SSD, (**c**) the DSD, (**d**) the SRSD, (**e**) the DRSD, (**f**) the SFSD, and (**g**) the LSD.

The randomization of support device distribution in the scene is another critical aspect influencing the extraction impact of the support devices. As illustrated in Figure 8, the distribution of support devices in all scenes falls into three categories: symmetric distribution (SD), asymmetric distribution (AD), and neighboring distribution (ND). The SD scene includes support devices such as a DSD and an SRSD. AD scenes mainly consist of DSDs. The support device is close together in an ND situation with the DSD and LSD.

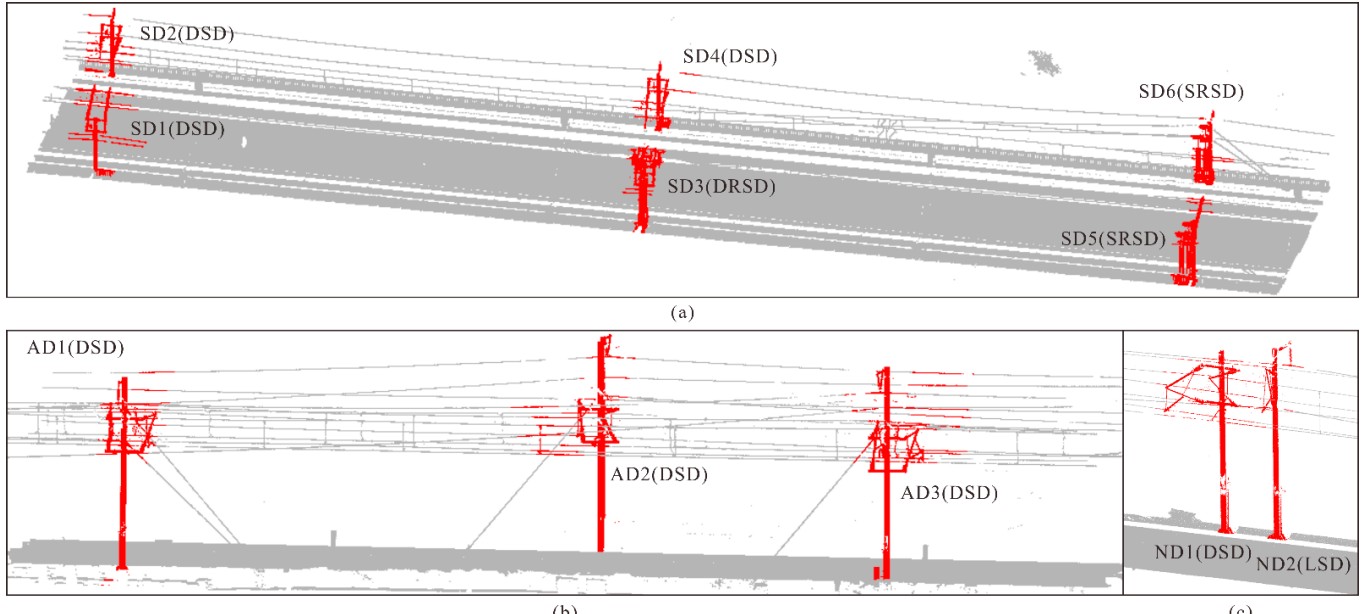

**Figure 8.** Display diagram of the three types of distribution of support device scenes: (**a**) the SD, (**b**) the AD, and (**c**) the ND.

### 3.2. Implemental Details

The experimental parameters of the suggested approach are shown in Table 1. Specifically, we analyzed the influence of the length of $RBox^{SD}$ and $RBox^{P}$ on the batch processing performance in the discussion and we set $RBox^{SD}$ and $RBox^{P}$ at 2 m and 3.5 m above the key trajectory point, respectively, through prior knowledge. Furthermore, to ensure that the $RBox^{N}$ contained as the whole of the pillar was feasible, its length and width should have been twice the pillar length and width, and its height should have been twice the $RBox^{P}$. While the $RBox^{PF}$ and $RBox^{PI}$ were built from the pillar's center point, their length and breadth only needed to be somewhat more significant than the pillar's length and width. In Section 2.4, to realize the effective separation of the support device and suspension insulator, we constructed the $RBox^{PLeft}$ and $RBox^{PRight}$ based on the pillar's center point, which were slightly longer and wider than those of the support device.

**Table 1.** Descriptions of the parameters.

| Parameter | Description | Value |
|---|---|---|
| $RBox^{SD}$ | support device region box (affine transformation is required) | length: 30 m, width: 22 m, height: 1.0 m |
| $RBox^{P}$ | pillar region box (affine transformation is required) | length: 30 m, width: 22 m, height: 3.5 m |
| $RBox^{N}$ | neighborhood region box | length: 0.8 m, width: 0.8 m, height: 2.0 m |
| $RBox^{PI}$ | pillar inspection region box | length: 0.4 m, width: 0.4 m, height: 3.5 m |
| $RBox^{PF}$ | pillar filter region box | length: 0.4 m, width: 2.6 m, height: 3.5 m |
| $RBox^{PLeft}$ | the left region box adjacent to $RBox^{PF}$ | length: 6.2 m, width: 3.0 m, height: 3.5 m |
| $RBox^{PRight}$ | the right region box adjacent to $RBox^{PF}$ | length: 6.2 m, width: 3.0 m, height: 3.5 m |
| $RBox^{IE}$ | the initial extraction region box of the support device | length: 12.8 m, width: 3.0 m, height: 3.5 m |
| $\psi$ | the rarefying threshold of trajectory data | 10 |
| $\delta$ | pillar center point initial extraction threshold in $RBox^{N}$ | 1200 |
| $\lambda$ | pillar center point check threshold in $RBox^{PI}$ | 2500 |
| $d$ | voxel width | $1/16 w_0$ |
| $\varepsilon$ | contact line rejection threshold in voxel | 25 |
| $w_0$ | voxel length | 0.06 m |

### 3.3. Evaluation Indexes

The algorithm's effectiveness, robustness, and practical application abilities are validated through quantitative and qualitative analysis and the application evaluation of the six support devices and three support device distribution scenes. The precision (P), recall (R), F1-score (F1), and intersection over union (IoU) were used to evaluate the extracted results. Here, the F1 and IoU represent the overall extraction effect, $P$ is the exact prediction result ($\boldsymbol{P}^{TPre}$) of the predicted result ($\boldsymbol{P}^{Pre}$), and it is used to assess the filtering effect of the contact wire, and R represents the proportion of $\boldsymbol{P}^{TPre}$ in the actual results ($\boldsymbol{P}^{Real}$), which is used to evaluate the extraction effect of the support device. The relevant formula is as follows:

$$P = \frac{\boldsymbol{P}^{TPre}}{\boldsymbol{P}^{Pre}} \tag{5}$$

$$R = \frac{\boldsymbol{P}^{TPre}}{\boldsymbol{P}^{Real}} \tag{6}$$

$$F1 = \frac{2 * P * R}{(P + R)} \tag{7}$$

$$IoU = \frac{\boldsymbol{P}^{Pre} \cap \boldsymbol{P}^{Real}}{\boldsymbol{P}^{Pre} \cup \boldsymbol{P}^{Real}} \qquad (8)$$

### 3.4. Experimental Results

The application tests reviewed two sets of 2 km railway datasets to further validate the suggested algorithm's applicability in the unit scenes. In units 1 and 2, there were 43 and 52 support devices, respectively, and two steel frame support devices were extracted wrongly in Figure 9b and one support device was removed in Figure 10c. The extraction issue occurs because these support devices do not require pillar erection for the algorithm to realize the support device's location, and the omission is due to the omission of the pillar center point. The mean IoUs of the correctly extracted support device in unit 1 and unit 2 were 95.85% and 96.17%, respectively. The results show that the algorithm has good application ability for a wide range of railway scenes.

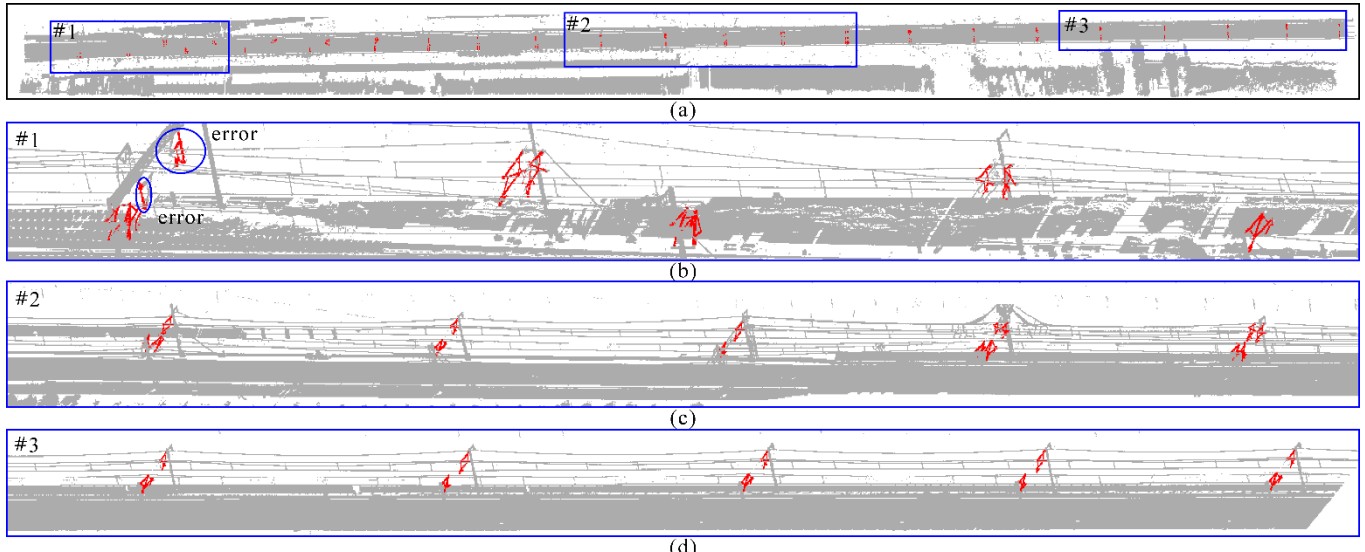

**Figure 9.** Applied experiment test results of unit 1. Specifically, (**a**) shows the test results, (**b**) shows close-up views of region #1, (**c**) shows close-up views of region #2, and (**d**) shows close-up views of region #3.

Table 2 displays the evaluation findings of the six different types of support devices. The mean values of F1 and IoU are 98.23% and 97.22%, respectively, and the F1 and IoU evaluation results of the double support device and the ratchet double support device are lower than the average. The reason is that a tiny number of insulators were misidentified as non-supporting devices and filtered away during initial extraction. The primary support device in the railway scene was the single support device with a reasonably simple structure. Therefore, the proposed method has the best extraction effect for a single support device. The extraction effect of the multi-type support device is shown in Figure 11.

**Table 2.** Extraction outcome quantitative evaluation.

| Types<br>Predict (%) | SSD | DSD | SRSD | DRSD | SFSD | LSD | Average |
|---|---|---|---|---|---|---|---|
| P (%) | 99.59 | 98.02 | 99.74 | 99.74 | 98.76 | 99.83 | 99.28 |
| R (%) | 97.53 | 97.97 | 97.70 | 96.52 | 98.94 | 97.38 | 97.67 |
| F1 (%) | 97.14 | 98 | 98.71 | 98.1 | 98.85 | 98.59 | 98.23 |
| IoU (%) | 98.55 | 96.08 | 97.46 | 96.28 | 97.73 | 97.23 | 97.22 |

(a)

#1

(b)

#2
omission

(c)

#3

(d)

**Figure 10.** Applied experiment test results of unit 2. Specifically, (**a**) shows the test results, (**b**) shows close-up views of region #1, (**c**) shows close-up views of region #2, and (**d**) shows close-up views of region #3.

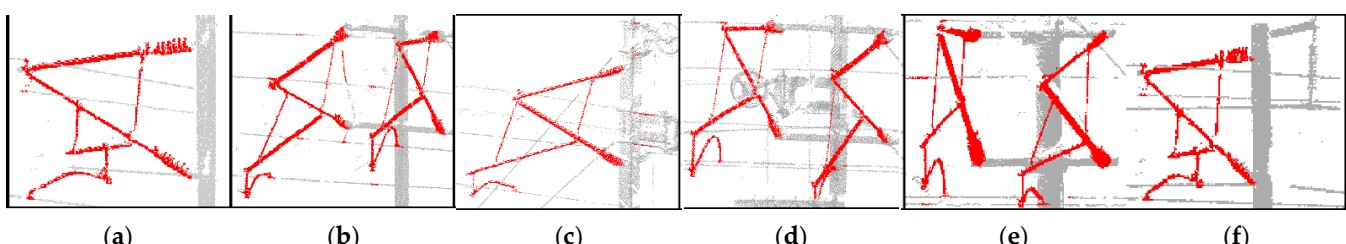

(**a**)      (**b**)      (**c**)      (**d**)      (**e**)      (**f**)

**Figure 11.** Extraction results of the six types of support devices: (**a**) the extracted result of the SSD, (**b**) the extracted result of the DSD, (**c**) the extracted result of the SRSD, (**d**) the extracted result of the DRSD, (**e**) the extracted result of the SFSD, and (**f**) the extracted result of the LSD.

Table 3 and Figure 12 show the extraction effects of the three support device distribution scenes. The average IoU and F1 of the symmetric distribution scene were 95.89% and 97.88%, respectively. The average IoU and F1 of the asymmetric distribution scene were 94.49% and 97.15%, respectively. The average IoU and F1 of the adjacent distribution scenes are 92.49% and 96.05%, respectively. Because some of the insulators are filtered out, the IoU and F1 of SD5, SD6, and AD2 are substantially lower than the average symmetric and asymmetric distribution accuracy. In the neighboring distribution scene, the IoU and F1 of ND2 are well below the average accuracy due to the unfiltered slings between the adjacent support devices.

**Table 3.** Quantitative extraction results.

| Scene | SD | | | | | | | AD | | | | ND | | |
|---|---|---|---|---|---|---|---|---|---|---|---|---|---|---|
| Predict | SD1 | SD2 | SD3 | SD4 | SD5 | SD6 | Average | AD1 | AD2 | AD3 | Average | ND1 | ND2 | Average |
| P (%) | 99.08 | 99.85 | 99.72 | 98.09 | 95.07 | 99.12 | 98.48 | 98.40 | 99.43 | 98.85 | 98.89 | 98.05 | 97.77 | 97.91 |
| R (%) | 99.13 | 98.18 | 96.75 | 99.78 | 95.64 | 93.84 | 97.21 | 97.87 | 93.35 | 95.31 | 95.51 | 98.56 | 90.12 | 94.34 |
| F1 (%) | 99.11 | 99.01 | 98.21 | 98.92 | 95.64 | 96.41 | 97.88 | 98.13 | 96.29 | 97.05 | 97.15 | 98.3 | 93.79 | 96.05 |
| IoU (%) | 98.24 | 98.04 | 96.49 | 97.88 | 91.64 | 93.07 | 95.89 | 96.34 | 92.85 | 94.28 | 94.49 | 96.67 | 88.31 | 92.49 |

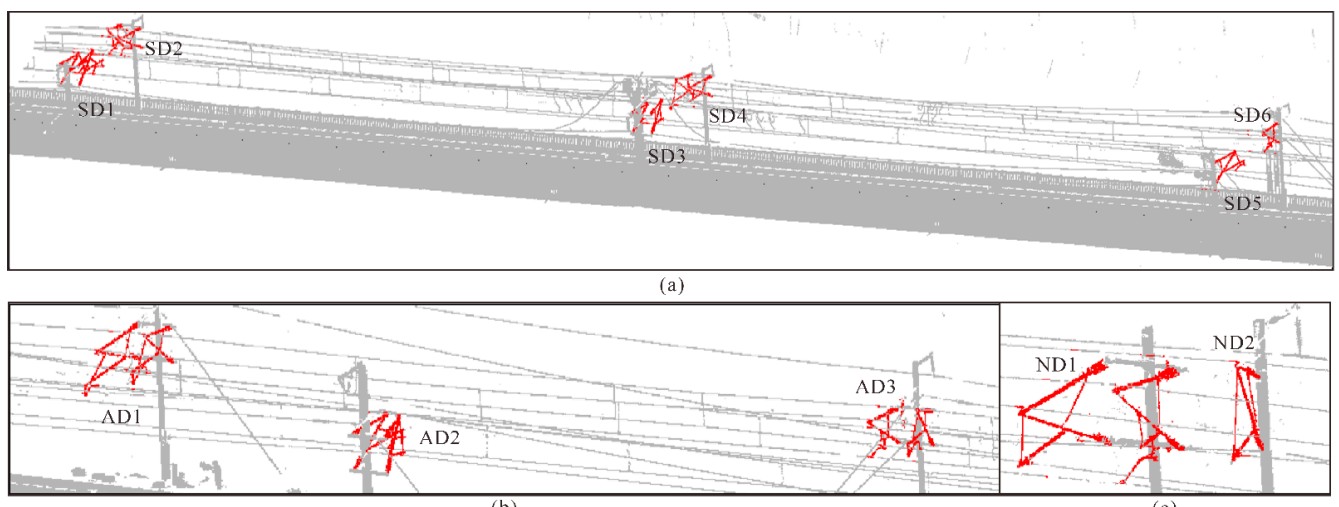

**Figure 12.** The test results of the support device scenes: (**a**) the SD extraction results, (**b**) the AD extraction results, and (**c**) the ND extraction results.

### 3.5. Ablation Experiments

In the process of result optimization, the filtering of other devices in the initial extraction results of the support device was realized by integrating the suspension insulator rough cutting (RC), pillar filter (PF), and voxel projection filtering (VPF). The impacts of pillars, suspension insulators, and contact wires were discussed through ablation tests on the final extraction results. The test results reveal that, while RC and VPF may remove most pillar point clouds, some pillar points remain in the black circle, as seen in Figure 13. Figure 14 shows that when the VPF method is not utilized, numerous contact wire point clouds in the black circle are not filtered out. Although the contact wire point cloud with a tiny scale of point cloud has a slight improvement on the extraction accuracy of the support device, it has a significant improvement on the visual effect, which is evident in the SRSD, DRSD, and DSD. Figure 15 shows the RC process for the suspension insulator device in the support device. The results reveal that the PF, RC, and VPF uniquely refine the support device's initial extraction results to increase the overall extraction accuracy.

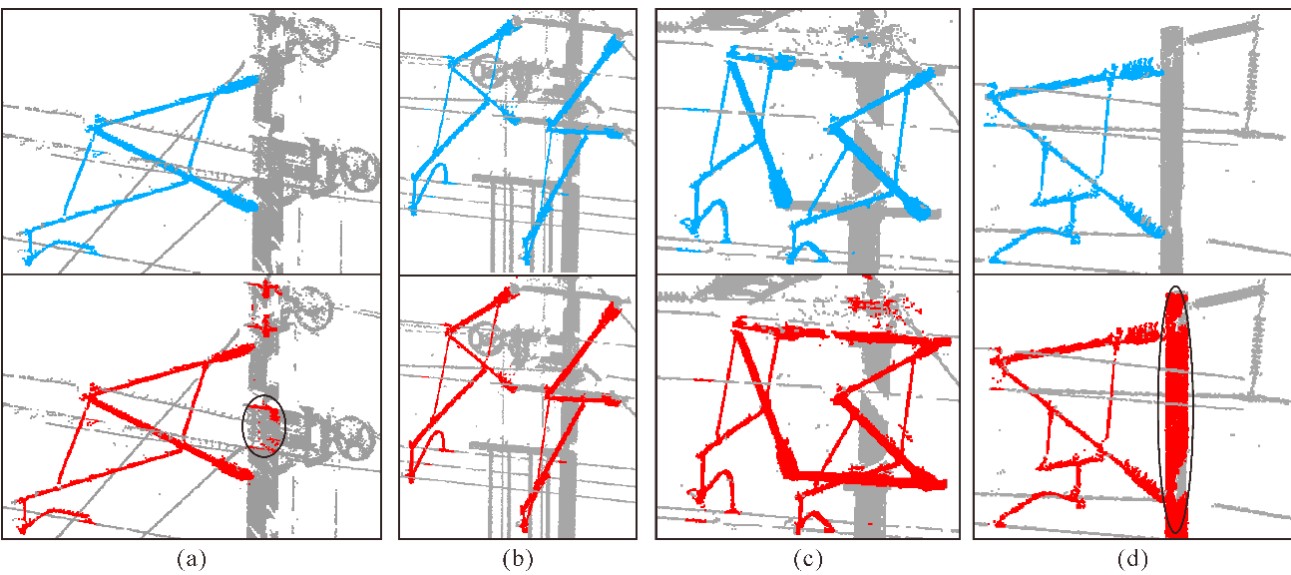

**Figure 13.** Differences in the extraction effects of the integrated PF, RC, and VPF filters and integrated RC and VPF filters. Specifically, the blue results are the extractions of the integrated PF, RC, and VPF, and the red results are the extractions of the integrated RC and VPF. (**a**) The application of the two filters on the SRSD. (**b**) The application of the two filters on the DRSD. (**c**) The application of the two filters on the SFSD. (**d**) The application of the two filters on the LSD.

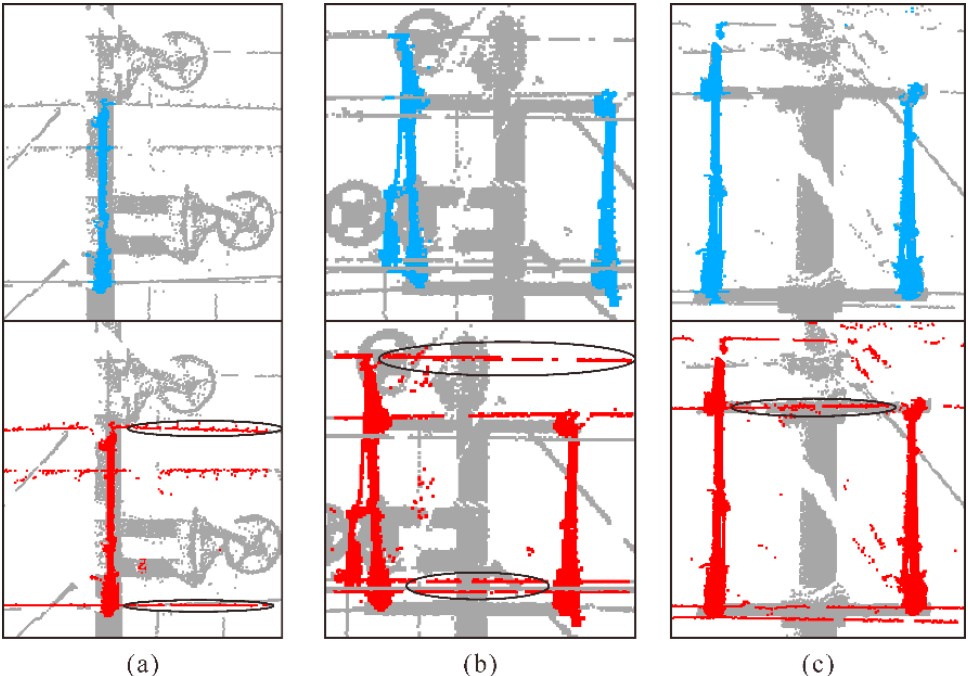

**Figure 14.** Differences in the extraction effects of the integrated PF, RC, and VPF filters and integrated PF and RC filters. Specifically, the blue results are the extractions of the integrated PF, RC, and VPF, and the red results are the extractions of the integrated PF and RC. (**a**) The application of the two filters on the SRSD. (**b**) The application of the two filters on the DRSD. (**c**) The application of the two filters on the SFSD.

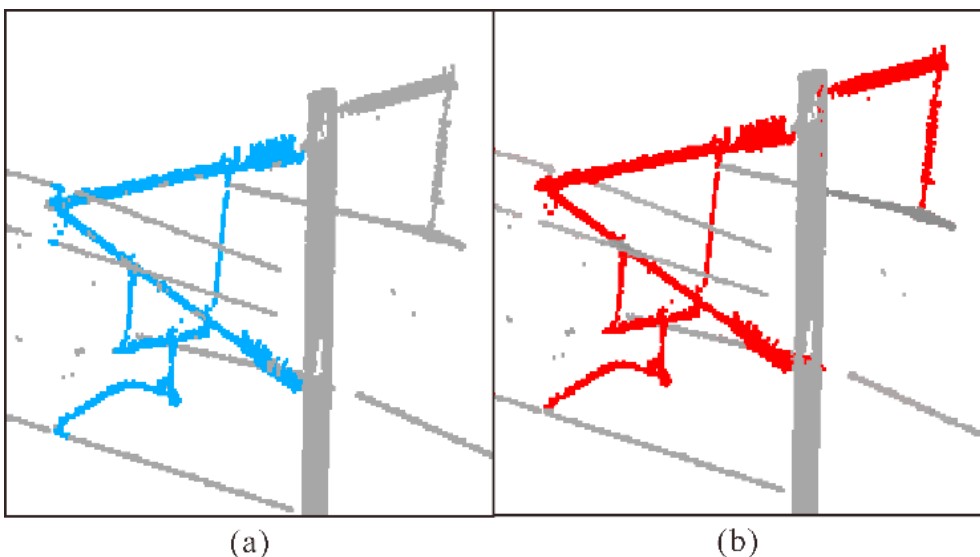

**Figure 15.** Different extraction effects of the integrated PF, RC, and VPF filters and integrated PF and VPF filters. Specifically, (**a**) is the extraction of the integrated PF, RC, and VPF and (**b**) is the extraction of the integrated PF and VPF.

## 4. Discussion

### 4.1. Analysis of Rarefying Threshold

The key trajectory points provide important spatial reference values for the attitude adjustment of the subsequent region blocks as a result of the trajectory point thinning. The spacing of the key trajectory point (*dis*) is strongly related to the thinning threshold ($\psi$) that affects the scale of $RBox^{SD}$ and $RBox^{P}$ in the hierarchical chunking process. The influence of the value of $\psi$ on the method's performance in symmetric distribution scenes is evaluated in this work. In an ideal railway arrangement, a single trajectory point should correspond to two support devices on either side of the track. To ensure the spatial reference value of the key trajectory points to the region block of the support device, the ratio between the key trajectory points and the support devices on both sides (*RKTO*) should be more than 0.5. The experimental results are shown in Table 4. The processor of the test equipment was a 12th Gen Intel(R) Core(TM) i7-12700H 2.70 GHz, and its onboard memory was 16.0 GB. As $\psi$ increases, the running time decreases, and so does the *dis* and the ratio. Therefore, given the actual requirements and algorithm time consumption, we set $\psi$ and *dis* at 10 and 40 m, respectively. In addition, there was a 4 m overlap area between the two adjacent $RBox^{SD}$ or $RBox^{P}$ to avoid the repeated extraction caused by hierarchical chunking. In a nutshell, the widths of $RBox^{SD}$ and $RBox^{P}$ were set to 22 m.

**Table 4.** Trajectory rarefying threshold test results.

| Parameter | Experimental Results | | | | | | | | | | |
|---|---|---|---|---|---|---|---|---|---|---|---|
| *n* | 5 | 6 | 7 | 8 | 9 | 10 | 11 | 12 | 13 | 14 | 15 |
| *dis* (m) | 20 | 24 | 28 | 32 | 36 | 40 | 44 | 48 | 52 | 56 | 60 |
| *RKTO* | 1 | 5/6 | 5/6 | 2/3 | 2/3 | 1/2 | 1/2 | 1/2 | 1/3 | 1/3 | 1/3 |
| Running time (s) | 1030 | 1010 | 990 | 990 | 980 | 980 | 980 | 970 | 960 | 960 | 940 |

### 4.2. Analysis of the Thresholds of the Pillar Center Points

The extraction threshold ($\delta$) of the pillar neighborhood search and the filtering threshold ($\lambda$) of the pillar center inspection area will significantly affect the extraction result of the pillar center. Therefore, we observed changes in the number of pillar center extraction results in the symmetric distribution scenes by adjusting the values of $\delta$ and $\lambda$ (Figure 16). There were six pillars in the original scene. The number of pillar center points extracted

decreased with the increase in parameters $\delta$ and $\lambda$ and the number of pillar center points extracted by the algorithm matched the number of pillars in [200,1600] and $\delta$ in [1250,3250]. This suggests that when $\delta$ and $\lambda$ are too low, the presence of compensation devices is misidentified as pillars and they not detected during inspection, resulting in more extracted results than the actual number of pillars. When $\delta$ and $\lambda$ are too high, some pillars are ignored during the neighborhood search, or some correct results are filtered during checking, resulting in fewer extracted pillars than the actual number of pillars. In addition, when $\delta$ is too low, we can use a larger $\lambda$ to filter the wrong results in the checking process, as shown in Figure 15. When $\delta$ is 100, $\lambda$ is 4250 or 4000, and the number of extracted pillars is correct. However, considering the applicability of the railway scene, relatively average parameters are chosen as $\delta$ and $\lambda$ in this paper, that is, $\delta$ is 900 and $\lambda$ is 2250.

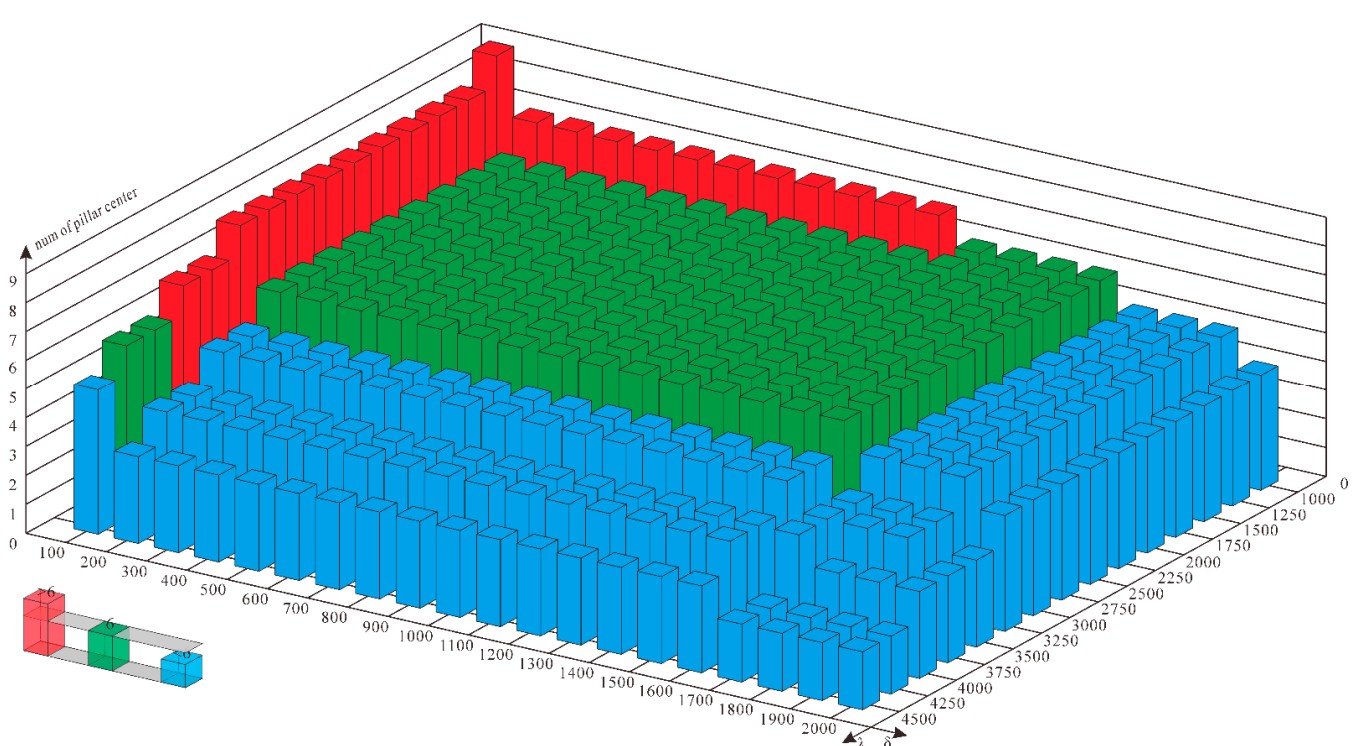

**Figure 16.** The relationship between the pillar center point extraction and the elimination threshold.

### 4.3. Analysis of Voxel Size and Contact Line Threshold

The number of support devices and the number of contact wire points in the voxel area are strongly related to the filtering effect of voxel projection, and the widths of the two ends of the flat wrist arm are noticeably different. As a result, multiple voxel area sizes were investigated in this research to determine the best voxel filtering threshold ($\varepsilon$), allowing for successful contact line filtering. At the same time, because of the vertical distribution between the contact wire and the support device, the *X*-axis edge ($l_0$) of the voxel area is longer than the *Y*-axis edge ($d$) to improve the extraction accuracy of the support device. Considering the above factors, we set the $l_0$ test interval to be $[0.03, 0.06] \cup [0.15, 0.18]$ and the $d$ test interval to be $[w_0/16, w_0]$, where $w_0$ is the length of the upper side of the coarse clipping area in the *Y*-axis direction. The test results are shown in Figure 17.

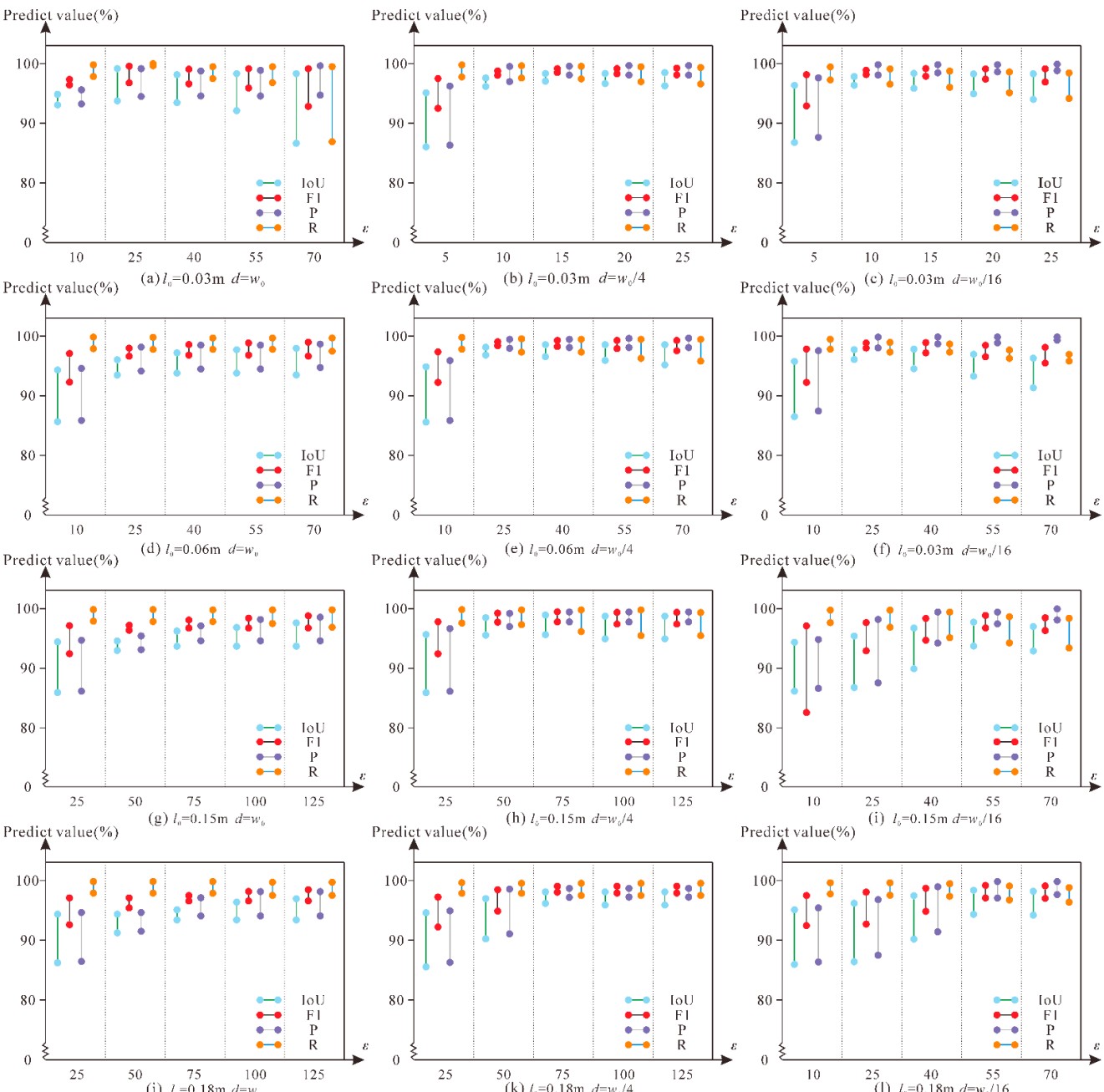

**Figure 17.** The relationship between voxel size and voxel filter threshold.

The upper end of the dumbbell seen in Figure 17 represents the highest test accuracy of the six types of support devices, while the lower end represents the lowest. When this parameter is used, the median length of the dumbbell reflects the algorithm's stability, and the longer the length, the larger the precision fluctuation and the lower the stability. As shown in Figure 17a, with the increase in $\varepsilon$, P gradually increases and R gradually decreases and the IoU and F1 first increase and then decrease. This shows that the contact wire and part of the support device point cloud are gradually filtered out, but the filtered part of the support device can be ignored, and so the IoU and F1 gradually increase. Then, as $\varepsilon$ becomes more prominent, most of the contact wires are filtered out, and the support device filter portion begins to affect the final extraction accuracy, leading to a gradual decrease in the IoU and F1. In addition, With the gradual decrease in $d$ and $w_0$, the points in the voxel region are gradually reduced and the overall stability of the algorithm is gradually

improved. In summary, the values of $w_0$, $d$, and $\varepsilon$ are closely related to the final extraction effect of the algorithm.

To more clearly show the relationship between the voxel scale and extraction effect, the optional $\varepsilon$ in each voxel is shown in Figure 18. With the increase in $w_0$, the optimal IoU and F1 gradually increase, but the disparity between the maximum and minimum of IoU and F1 increases significantly. With the increase in $d$, the disparity between the maximums and minimums of the IoU and F1 only increase at first, and then they decrease. Therefore, $w_0$ is set to 0.06, $d$ is $1/16w_0$, and the optimal $\varepsilon$ is 15.

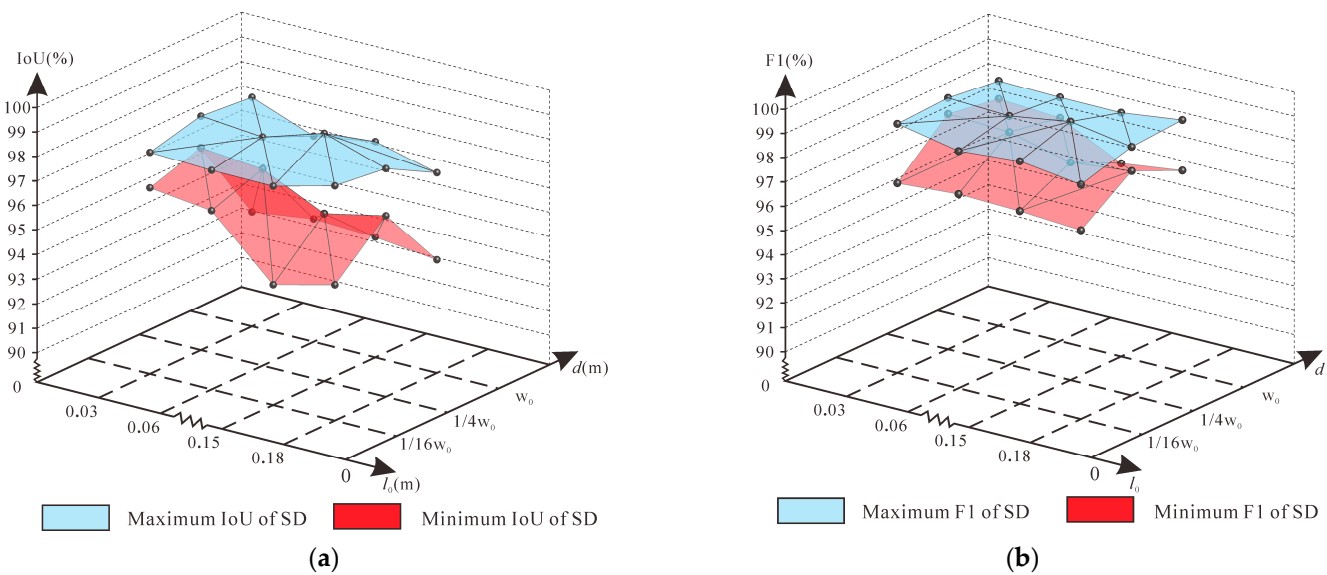

**Figure 18.** The extraction accuracy corresponds to each voxel. (**a**) The IoU corresponding to each voxel. (**b**) The F1 corresponding to each voxel.

### 4.4. Discussion on Point Sparsity

The point sparsity of small-scale targets is mainly determined by their size and the scanning point density of the laser scanning equipment. In this manuscript, the proposed method can break through the size limitation of the support device itself and achieve the fast and efficient extraction of the support devices in complex railway datasets. In addition, the core of the proposed algorithm is the relative spatial relationship between railway devices in the railway scene. Therefore, the decline of the overall point density has a limited impact on the processing performance of the algorithm. To prove this argument, we diluted the point cloud data of the original six support devices to one-fifth of the original so as to test the performance of the proposed algorithm on sparse point clouds. The test results are shown in Table 5 and Figure 19. The accuracy of the test results is more sensitive because of the sparse points, and the extraction accuracy is lower than the original data. However, after thinning by 4/5 of the data, the mean F1 and IoU of the six support devices can still achieve 97.29% and 94.75%, respectively, demonstrating that the proposed method can still complete the automatic and accurate extraction of the support device in the railway dataset. Thus, the proposed method can still realize the automatic extraction of support devices from railway data with low-density point clouds.

**Table 5.** Test results.

| Type<br>Predict | SSD | DSD | SRSD | DRSD | SFSD | LSD | Average |
|---|---|---|---|---|---|---|---|
| Origin points number | 105,112 | 270,310 | 454,728 | 835,883 | 995,035 | 106,517 | 461,264 |
| Filtered points number | 21,023 | 54,062 | 90,946 | 167,177 | 199,007 | 21,304 | 92,253 |
| Origin P (%) | 99.59 | 98.02 | 99.74 | 99.74 | 98.76 | 99.83 | 99.28 |
| P (%) | 97.93 | 96.48 | 99.51 | 99.40 | 99.16 | 97.01 | 98.25 |
| Origin R (%) | 97.53 | 97.97 | 97.70 | 96.52 | 98.94 | 97.38 | 97.67 |
| R (%) | 95.43 | 98.16 | 94.61 | 96.63 | 97.98 | 94.69 | 96.25 |
| Origin F1 (%) | 97.14 | 98.00 | 98.71 | 98.1 | 98.85 | 98.59 | 98.23 |
| F1 (%) | 96.66 | 97.31 | 97.01 | 97.99 | 98.57 | 96.22 | 97.29 |
| Origin IoU (%) | 98.55 | 96.08 | 97.46 | 96.28 | 97.73 | 97.23 | 97.22 |
| IoU (%) | 93.55 | 94.77 | 94.18 | 96.07 | 97.18 | 92.72 | 94.75 |

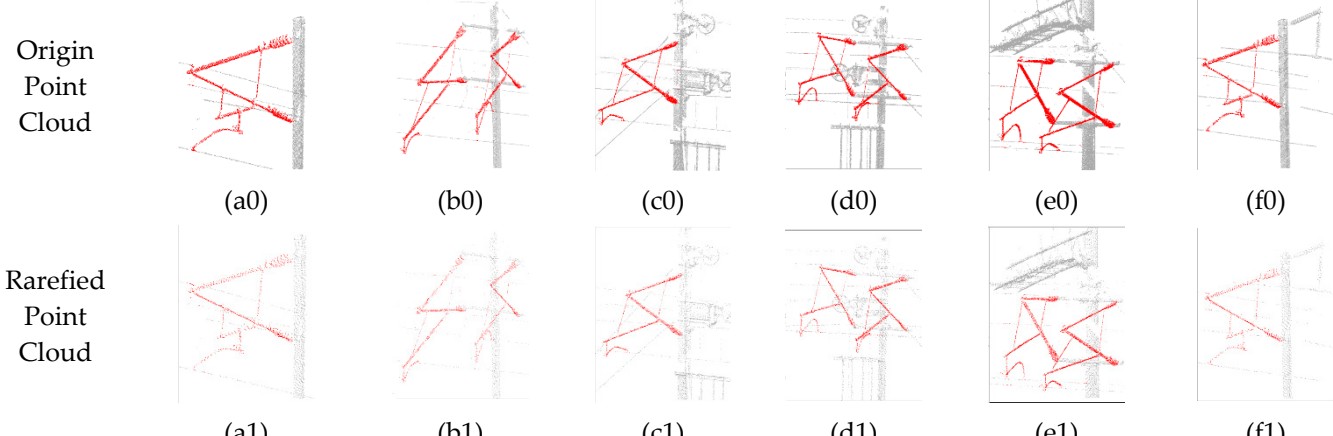

Origin
Point
Cloud

(a0)  (b0)  (c0)  (d0)  (e0)  (f0)

Rarefied
Point
Cloud

(a1)  (b1)  (c1)  (d1)  (e1)  (f1)

**Figure 19.** Comparison of the point cloud data before and after thinning. (**a0**) is the processed result of origin point cloud of SSD, (**a1**) is the processed result of rarefied point cloud of SSD, (**b0**) is the processed result of origin point cloud of DSD, (**b1**) is the processed result of rarefied point cloud of DSD, (**c0**) is the processed result of origin point cloud of SRSD, (**c1**) is the processed result of rarefied point cloud of SRSD, (**d0**) is the processed result of origin point cloud of DRSD, (**d1**) is the processed result of rarefied point cloud of DRSD, (**e0**) is the processed result of origin point cloud of SFSD, (**e1**) is the processed result of rarefied point cloud of SFSD, (**f0**) is the processed result of origin point cloud of LSD, (**f1**) is the processed result of rarefied point cloud of LSD.

## 5. Conclusions

As the core device of a railway system carrying the power supply system, the operation status of a support device is directly related to the normal operation of a railway system. At present, the acquisition of information related to support devices is still primarily manual. MLS technology can efficiently obtain geometric and spectral information about surrounding objects and is not affected by external factors such as lighting conditions. This is very important for the routine inspection of a railway, which is often carried out at night. However, the processing method of massive point clouds in large scenes is not mature. Therefore, to quickly and effectively extract the point cloud data of support devices for routine railway inspection, such as defect detection and geometric parameter measurement, an automatic extraction method of support devices based on the MLS point cloud is proposed in this paper. First, the large-span railway scene is batch processed by the hierarchical chunking the railway scene data. Second, the positioning and initial extraction of the support device are realized based on the reasonably steady spatial relationship between the pillar and the supporting device in the scene. The initial extraction results are then optimized by the integrated pillar filter and voxel projection filter to realize high-precision support device extraction in complicated railway scenarios. Furthermore, the

algorithm's relative parameters are verified and analyzed in the discussion section, and the automatic extraction of assistance devices in railway scenarios can be realized using the analyzed parameters. We tested the algorithm's performance with six different types of support devices and three different distribution scenes. Test samples were obtained from two railway track data sets using the Z + Fprofile 9012 laser scanning equipment. The quantitative and qualitative test results revealed that the extraction support device's P, R, F1, and IoU coefficients may reach more than 95%, and the overall visual effect of the extraction results is good. Furthermore, the average mean IoU of 96.11% and the good visual effect in the application test of the two groups of 2 km railway units demonstrated that the method has significant robustness.

**Author Contributions:** Conceptualization, S.Z. and Q.M.; methodology, S.Z.; software, S.Z.; validation, S.Z., Y.H. and Z.F.; formal analysis, L.C.; investigation, S.Z.; resources, S.Z.; data curation, S.Z.; writing—original draft preparation, S.Z.; writing—review and editing, S.Z.; visualization, S.Z.; supervision, S.Z.; project administration, S.Z.; funding acquisition, S.Z. All authors have read and agreed to the published version of the manuscript.

**Funding:** This research received no external funding.

**Conflicts of Interest:** The authors declare no conflict of interest.

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
