# Peer review of "A Method for the Automatic Extraction of Support Devices in an Overhead Catenary System Based on MLS Point Clouds"

_remotesensing, doi:10.3390/rs14235915_

Round 1
Reviewer 1 Report
Please explain the abbreviations at first occurrence in the text.
Page 5, chapter 2.3: The first sentence appears to be incomplete, 2nd and 3rd line – „... relationship between the support device and support device ...“, is it correct? May be the authors meant „... relationship between the pillar and support device ...“
Page 7, last line: Instead „... acenes ...“, may be „... scenes ...“
Page 8, chapter 3.2: first sentence „... approach are shown in ...“, there is missing number of figure or table.
Page 10 and 11: Please separate the figure caption and table caption by blank line, both texts are blending.
Reviewer 2 Report
This paper proposed an automatic extraction method for weak feature objects (support devices) in large-scale MLS point clouds of the overhead catenary system. The paper provided a complete introduction of the research background as well as the significance of the work and described the implementation process of the automatic extraction method in detail. The effectiveness and robustness of the proposed method are verified through extraction experiments for six different types of support devices in two real scanning point clouds.
In addition, I would like to know, in large-scale point clouds, due to the limitation of scanning devices, some small-scale targets often have the problem of sparse points or incomplete shapes, can the proposed method achieve the same high-precision automatic extraction on such data?
Overall, I think this work is very interesting and suitable for publication in the Remote Sensing journal.
Reviewer 3 Report
General comments
The manuscript could be considered for publication, but not in its present form. At least a major revision is essential, even if my comments are compatible with a "Reject & Resubmit" recommendation.
The authors have done an interesting job, but not intended for an adequate audience for a remote sensing journal if certain aspects are not addressed (please see these general comments, as well as the specific comments in the annotated PDF). The manuscript seems to me to be incomplete; the reader may wonder "So what? What do we do with this?"
It is not clear what the ultimate purpose of this study is, also because it is not well explained how the point clouds are acquired and therefore what the final data are. For example, is a support viewed only from one side and at a certain angle, or from two sides to reduce shadows? What are the geometric characteristics that can be used to evaluate the conditions of the support, and how is it guaranteed that, given the expected quality of the point cloud, these characteristics are really evaluable? What can actually be achieved for diagnostic purposes?
Moreover, a reader might argue that photogrammetric techniques (for 3D modeling) and semantic segmentation techniques (applied to some of the images used for photogrammetry) could be used more easily for the same purposes. To this objection it is easy to answer that inspections of a line are generally nocturnal (artificial lighting does not lend itself well to photogrammetry), but this fact must be said to avoid such an objection.
In conclusion, it is not clear what is the importance of the proposed method for a remote sensing audience or even for the railway engineering community. At the moment it almost feels like an academic exercise.
To make this incomplete manuscript a strong and complete article, it is necessary to characterize well what the authors want to achieve and show, with at least an example, what can (and what cannot) be obtained for diagnostic purposes, on the basis of the type of degradation that is expected for a support.
The observation of the insulators leads me to believe that the supply voltage is 25 kV a.c. 50 Hz (or 2x25 kV), but this important detail is omitted in the manuscript.
Another problem is that the article is sloppily prepared, it looks more like a draft than a submitted manuscript. For example, the size of the equations varies from line to line, as if the equation editor was used improperly (I still point out that with the .docx format the equation editor is not suitable and it is better to use Office resources for equation management).
Mathematical symbols are sometimes misused (see specific comments).
In general, the Instructions for Authors are disregarded.
Specific comments
Please see the annotated manuscript.

Round 2
Reviewer 3 Report
The authors significantly improved the manuscript and responded to all my comments acceptably.
The manuscript is now ready for publication and therefore I recommend its acceptance.
The authors improve the manuscript a lot and responded to all my comments acceptably. The manuscript is now ready for publication and therefore I recommend its acceptance. The authors improve the manuscript a lot and responded to all my comments acceptably. The manuscript is now ready for publication and therefore I recommend its acceptance.